# EMG and Joint Angle-Based Machine Learning to Predict Future Joint Angles at the Knee

**DOI:** 10.3390/s21113622

**Published:** 2021-05-22

**Authors:** Jordan Coker, Howard Chen, Mark C. Schall, Sean Gallagher, Michael Zabala

**Affiliations:** 1Department of Mechanical Engineering, Auburn University, Auburn, AL 36849, USA; jcc0041@auburn.edu (J.C.); howard-chen@auburn.edu (H.C.); 2Department of Industrial Engineering, Auburn University, Auburn, AL 36849, USA; mcs0084@auburn.edu (M.C.S.J.); seangallagher@auburn.edu (S.G.)

**Keywords:** EMG, prediction, machine learning, joint angle

## Abstract

Electromyography (EMG) is commonly used to measure electrical activity of the skeletal muscles. As exoskeleton technology advances, these signals may be used to predict human intent for control purposes. This study used an artificial neural network trained and tested with knee flexion angles and knee muscle EMG signals to predict knee flexion angles during gait at 50, 100, 150, and 200 ms into the future. The hypothesis of this study was that the algorithm’s prediction accuracy would only be affected by time into the future, not subject, gender or side, and that as time into the future increased, the prediction accuracy would decrease. A secondary hypothesis was that as the number of algorithm training trials increased, the prediction accuracy of the artificial neural network (ANN) would increase. The results of this study indicate that only time into the future affected the accuracy of knee flexion angle prediction (*p* < 0.001), whereby greater time resulted in reduced accuracy (0.68 to 4.62 degrees root mean square error (RMSE) from 50 to 200 ms). Additionally, increased number of training trials resulted in increased angle prediction accuracy.

## 1. Introduction

Active exoskeletons are devices that augment the performance of the wearer by providing mechanical assistance using powered actuators [1]. These devices are useful for rehabilitation as well as for pushing able-bodied individuals beyond their natural abilities [2]. Metabolic cost is often considered the gold standard for assessing lower-limb exoskeleton performance, given that the overall goal is to improve walking and running economy [3]. The first exoskeleton known to break the metabolic cost barrier was reported by Malcolm et al. (2013), which improved walking economy by 6% by using a tethered pneumatically-actuated ankle exoskeleton on a treadmill [4]. Mooney et al. (2014) then reported a 10% metabolic cost reduction with walking using an untethered electrically-actuated ankle exoskeleton on a treadmill [5]. To date, over 20 studies have demonstrated exoskeleton designs that have improved human walking and running economy through reductions in metabolic costs [3]. It is without doubt that these studies have thoroughly advanced the development of exoskeletons. However, continued development into exoskeleton technologies is necessary to reach practicality.

Exoskeleton control is an area of considerable research, given its importance for metabolic cost reductions and naturalistic operation. In general, an exoskeleton uses various controllers to map the user’s intent to signals that subsequently drive the actuators on the exoskeleton. A hierarchical control strategy consisting of high-level, mid-level, and low-level controllers is often used [6]. The high-level controller is used to perceive the environment and the user’s locomotive intent, such as standing/walking [7,8], estimation of incline slopes [9,10,11,12,13], and stair ascent/descent [10,13,14]. Information inferred by the high-level controller is used to by the mid-level controller, which subsequently translates the user’s motion intentions to desired device states (i.e., joint positions, velocities, torque). Various mid-level controllers determine device states based on gait phases. The simplest mid-level controllers that consider gait phase would time the actuators based on heel strike [4,5,15] or toe-off [16] determined using a footswitch. Recent approaches can estimate gait phase percentages from sensor data in a variety of environments [13,17], which can subsequently be used to help to optimize exoskeleton actuation to minimize metabolic rate for a given subject [18]. In lieu of leveraging gait phase information, actuation information may be inferred based on movements of other body segments [19].

High- and mid-level controllers can infer user intent and track device states using various sensors including inertial measurement units [20,21,22,23], rotary encoders [24], and electromyography (EMG) [9,25]. EMG measures electrical activity produced by muscle cells during muscle activation [26], and is commonly used in various biomedical applications [27] such as occupational ergonomics [28], rehabilitation [29,30], and nutrition [31]. This measurement modality is of particular interest towards naturalistic exoskeleton control since muscle activation is visible about 100 ms before muscle movement occurs, thereby able to ‘predict’ human motion [26], which can potentially further reduce metabolic costs. High-level controllers have inferred discrete actions using EMG measurements, with classification algorithms [10,32] that often require features [6], or alternative, wavelets [33] to be pre-computed. Mid-level controllers map EMG measurements into a continuous signal that is subsequently executed by the low-level controller using regression algorithms. This can be accomplished using proportional control [25], which multiplies the smoothed and rectified EMG signal by a constant gain factor as the desired velocity or torque [26]. Alternatively, biomechanical models can be used [34,35,36] to achieve higher fidelity. However, given the myriad parameters that must be determined, many researchers prefer using a model-free approach by leverage machining learning. 

Mapping EMG measurements to joint positions is a common approach for a regression-based EMG control scheme for exoskeleton control [37,38,39,40,41]. Lee and Lee (2005) estimated knee angles from EMG measurements using a combination of a radial basis function neural network and a multilayer neural network [41]. Aung et al. (2012) estimated shoulder flexion/extension and abduction/adduction angles from EMG measurements using a backpropagation neural network [37] Zhang et al. (2012) estimated knee, hip, and knee joint angles using a back-propagation neural network [40]. Chen et al. (2018) used a deep belief network that consists of restricted Boltzmann machines to estimate flexion/extension angles of the hip, knee, and ankle [38]. More recently, deep recurrent neural network [42], transfer learning [43], random forest [44], and genetic algorithms [45] have been proposed to estimate joint kinematics of the knee using EMG. 

Few studies, however, have leveraged the time shift between the registration of muscle activity to when movement occurs. Ma et al. (2020) considered time advancing the EMG signal to increase estimation accuracy [46]. To our knowledge, no studies have investigated the relationship between estimation accuracy and prediction horizon of EMG signals (e.g., the extent to which EMG signals can be time advanced). Understanding this relationship may enable the use of more accurate but computationally expensive joint estimation and/or low-level control methods (e.g., model predictive control) to be effectively used to control exoskeletons in real time. 

Therefore, the purpose of this study was to assess the performance of a supervised learning artificial neural network (ANN) algorithm trained with both knee flexion angle and knee muscle EMG signals during walking to predict knee flexion angle during walking at various amounts of time into the future. The primary hypothesis of this study was that the algorithm’s knee flexion angle prediction accuracy would only be affected by time into the future and that as time into the future increased, the prediction accuracy would decrease. A secondary hypothesis was that as the number of algorithm training trials increased, the prediction accuracy of the ANN would increase.

## 2. Materials and Methods

Ten subjects participated in this study (5 males and 5 females, age = 21.5 ± 2.0 yrs, weight = 64.5 ± 9.8 kg, and height = 166.9 ± 14.5 cm). Subjects reported no history of chronic pain in the spine or lower extremities in the six months prior to participating in this study. All study procedures were approved by the Auburn University Institutional Review Board (IRB), and subjects provided informed written consent before participating. The experiment took place at the Auburn University Biomechanical Engineering (AUBE) Laboratory.

Twelve surface EMG electrodes (Delsys Trigno IM, Delsys Inc., Boston, MA, USA) were placed bilaterally on six thigh muscles. These muscles were the right and left tensor fasciae latae, rectus femoris, vastus medialis, vastus lateralis, biceps femoris, and semitendinosus (Figure 1). The placement of these electrodes was consistent with Seniam guidelines [47]. Before placing the electrodes, excessive hair was removed with a small electric hair trimmer and the skin was cleansed of oils and debris with an alcohol swab to improve the quality of the recorded signals. A double-sided adhesive was used to secure the sensors and sports wrap was added over the electrode to help prevent loss of signal connection. Before testing, the subjects were allowed time to acclimate to the equipment. Raw EMG signals were collected at 1111 Hz (the maximum capacity of the Delsys Trigno IM sensor) for each of the twelve channels and fed through a Butterworth filter to remove motion artifacts (<20 Hz) and high frequency aliasing effects (>500 Hz). These signals were then detrended and rectified so that the RMS for each muscle could be calculated.

A ten-camera Vicon motion capture system was used to track a 79 retroreflective marker set consistent with the work of Andriacchi et al. [48]. The cameras were equally spaced on the walls around the circumference of the AUBE Lab and at varying heights and directed such that the capture volume was located at the center of the room. A Vicon Lock+ box was used to ensure synchronicity between the Vicon motion capture and the Delsys Trigno sensor signals. Nexus software was used to collect the marker position at 120 Hz (Version 2.6.1; Vicon Motion Systems Ltd., Oxford Industrial Park, Oxford, UK). Marker positional data were transferred to Visual3D, where they were filtered with the default 15 Hz low-pass Butterworth filter to remove noise. Body segments were created using the marker positions following the International Society of Biomechanics recommendations [49]. Grood and Stunay’s joint coordinate system was used to calculate knee flexion [50]. The Visual 3D model of the knee provided all six degrees of freedom, but knee flexion angle was the only parameter utilized for this study.

Subjects performed 15 walking trials over a distance of approximately twenty feet at a self-selected pace. Minimal feedback was given in order to capture naturalistic movements. Collected trials were split into training and testing categories and these categories were chosen to counter learning effects. Ten trials were chosen to be treated only as training trials. The ten training trials were trials one, two, four, five, seven, eight, ten, eleven, thirteen, and fourteen. The five remaining trials were used to only to test the algorithm accuracy. This study used the time domain feature for analysis, which is the least computationally expensive feature in order to decrease the signal analysis time. MATLAB was used to create Nonlinear Input-Output Time Series Neural Network algorithms trained using Bayesian Regularization with a single hidden layer of ten nodes and a feedback delay set to two. The seven input variables for the algorithms were the six EMG signals on a single leg and the knee flexion angle calculated post hoc with Visual 3D. The algorithms output predictions of that same knee’s flexion angle estimated at 50, 100, 150, and 200 ms into the future. 

Ten algorithms were trained with between 1 and 10 trials for all four prediction times. The specific trials used for each number of training trials (1–10) was randomized (i.e., one trial may have used Trial 8, two trials may have used Trials 1 and 4, etc., Figure 2) as to prevent any learning effects. This was conducted for both right and left legs for all ten subjects. This resulted in a total of 800 trained algorithms (10 subjects × 2 legs × 10 trials × 4 times). Root mean square error (RMSE) was calculated by comparing the algorithms’ output angle against the motion capture-based calculation of the knee flexion angle for each data point of that subject’s five testing trials. The averages and standard deviations of RMSE for a subject’s five testing trials were calculated for each algorithm for each prediction time and each number of training trials. This resulted in a total of 80 average RMSE values (average of 5 testing trials for 10 subjects × 2 legs × 4 times).

Analysis of variance (ANOVA) of average RMSE values was used to test for the significance of each prediction variable (subject, gender, leg, and time) and their interactions. Time was treated as a repeated measures variable. The unadjusted algorithm error data were found to have fan-shaped residuals (indicating unequal variances by time), thus a log transformation of the algorithm predictions was used for the ANOVA. This resolved the equal variance assumption violation. Tukey Honestly Significant Difference (HSD) post hoc tests were used to evaluate significant differences between conditions for main effects or interactions that were significant. A logarithmic regression was used to develop a prediction model for error based on variables identified as significant in the ANOVA model. The type I error rate (alpha) was set at 0.05 for all tests.

## 3. Results

ANOVA results demonstrated that the main effect of time was statistically significant with respect to algorithm predictions (F3,48 = 307.07, *p* < 0.001). Table 1 provides the results of the Tukey HSD post hoc tests with a calculated Standard Error for Comparison of 0.07, Critical Q Value of 3.76, and Critical Value for Comparison of 0.18. Tukey HSD results demonstrated that algorithm predictions for each time period were significantly different from one another (Table 2).

ANOVA results revealed that subject, leg, and gender differences were not statistically significant in terms of the algorithm predictions, nor were any interactions among these variables. Thus, the final model indicates that the only significant factor relating to algorithm predictions was time into the future.

Figure 3 shows the regression model with a slope = 5.43 log(RMSE)/s, *p* < 0.001 and *y*-intercept = −0.344 log(RMSE), *p* < 0.001. The adjusted R^2^ fit was 0.77. Figure 4 shows the same data without logarithmic transformation, and Equation (1) shows the corresponding equation and how to estimate RMSE for with prediction time as the variable.

Figure 5 and Figure 6 show the average RMSE of all the subjects at various prediction times and the various number of training trials separated by the right and left leg. Both legs saw a decrease in the prediction error as the number of training trials increased and as the prediction times were decreased. The reduction in RMSE from a single training trial to ten training trials for all times and for both legs was fairly consistent with a mean reduction of 52.9 ± 4.0%. Figure 7 and Figure 8 show the standard deviation of the RMSE generally decreased with an increase in the number of training trials, although some exceptions can be seen such as the left leg at both 150 and 200 ms.
RMSE = 10^(5.43 * time − 0.344)^(1)

Figure 9 shows a box and whisker plot of the RMSE for each of the subjects’ algorithms trained with one (top) as well as all ten (bottom) available training trials. A similar trend is seen in both the right and the left leg, with the error and variation of predictions increasing the further into the future the algorithm attempted to predict. The median errors for both the right and left leg were reduced 83.7% (4.34 to 0.71 degrees RMSE) and 83.1% (3.82 to 0.64 degrees RMSE), respectively, when comparing 200 to 50 ms at ten training trials.

## 4. Discussion

This study investigated the accuracy of an ANN designed to predict future flexion angles of the knee using EMG measurements and past measurements of knee flexion. Our root mean square (RMS) errors ranged from <1° to >4°, depending on prediction horizon. To our knowledge, there are no studies that are directly comparable, given our use of past measurements of knee flexion angles in addition to EMG measurements. However, the results of our study are consistent with recent studies that used EMG to predict joint kinematics of the knee. Li et al. (2019) reported root mean square RMS errors around 5° using random forest with principal component analysis. Chen et al. (2018) reported RMS errors <4° using a deep belief network [38]. Ma et al. (2020) reported RMS errors <3.5° using a long short-term network with time-advanced features [46]. It is expected that our prediction is more accurate, given that we leveraged previous measurement of knee flexion angles as an additional source of information.

The primary hypothesis was supported—the algorithm’s knee flexion angle prediction accuracy was only affected by time into the future and as time into the future increased, the prediction accuracy decreased (Table 1, Figure 3 and Figure 4). The secondary hypothesis was also supported in that as the number of algorithm training trials increased, the prediction accuracy of the ANN increased (Figure 5 and Figure 6).

Previously reported delays of approximately 100 ms between EMG signal readability and ensuing motion (Novak and Reiner) provide useful context for the results of this study [26]. The logarithmic equation derived from Equation (1) indicates that at 100 ms, approximately 1.58 degrees of RMSE is to be expected. Moreover, all values of average RMSE in this study at 100 ms into the future have a RMSE less than 5° (2.12 ± 0.69 degrees, minimum: 1.27°, maximum: 4.00°). Although these values seem small, the acceptable amount of error of an actively driven lower-limb exoskeleton while being worn by a user remains unknown and should be tested empirically. Equation (1) predicts an RMSE of 0.85° at 50 ms into the future, which is an improvement from 100 ms; however, computational speed limitations will undoubtedly determine the utility of values predicted a mere 50 ms into the future for actively controlled lower-limb exoskeletons.

The results of this study also demonstrate the benefit of increased training trials on the predictive algorithm accuracy. Figure 3 and Figure 4 indicate diminishing returns from added training trials. The greatest benefit is evidently between one and six training trials. Conversely, there is not a substantial change in prediction accuracy for all times into the future and for both legs between six and ten training trials. This information will be helpful in future studies by allowing researchers to focus less on large numbers of training trials for improved accuracy and instead on other potential areas of improvement such as the parameters and type of predictive algorithm used and EMG sensor signal quality.

Future work beyond the scope of the current study may include introducing transitional movements into the training sets, or using techniques such as hyper-parameter tuning in order to design the optimal machine learning algorithm parameters for the desired prediction time. Further work should also explore whether a single algorithm can be utilized across an independent population. With the inclusion of multiple subjects or both of an individual’s legs into a single algorithm, various patterns can be learned in hopes of making predictions on a subject without the need for their specific individual training data. Moreover, the finding that time is the only significant factor affecting prediction accuracy opens the possibility to explore opportunities of using independent sets of data for training and testing such as inter-subject and inter-leg training and testing.

There are several limitations in this study. We used a relatively benign machine learning approach to reduce computation expense, given that our primary object was not necessarily to produce the highest prediction accuracy, but rather to understand the capability of machine learning algorithms to predict knee joint angles into the future. We hypothesize that using state-of-the-art techniques such as deep learning approaches will increase our prediction accuracy. The small sample size and narrow demographics of the subjects present another limitation. Age, percentage body fat, and activity level could impact the quality of predictions due to their impact on EMG signal quality. Additionally, the subjects engaged in an established gait cycle. Expanding the predictions to less repeatable actions will most likely cause the accuracies to decrease if using the same machine learning method. Further testing should be conducted to determine whether a single walking-based regression algorithm is sufficient for multiple actions. Furthermore, an area for debate is whether EMG signals should be mapped to joint angles, joint moments, joint velocities, or some other control signal for exoskeleton control [6].

## 5. Conclusions

This study successfully demonstrated the feasibility of employing an ANN to predict knee flexion angle within a reasonable degree of accuracy. The results of this study show that only time into the future, not subject, gender, or side, affects algorithm prediction accuracy. The results also provide the basis for understanding the amount of data needed to train an angle-predicting algorithm for a given subject as greater than six training trials added no apparent benefit to predictive accuracy. Although, it remains to be seen how acceptable these amounts of error are for active lower-limb exoskeleton control and operator use, the results from predictive algorithms such as what is presented in this study will benefit powered exoskeleton control design and potentially ultimately result in improved user–machine interaction.

## Figures and Tables

**Figure 1 sensors-21-03622-f001:**
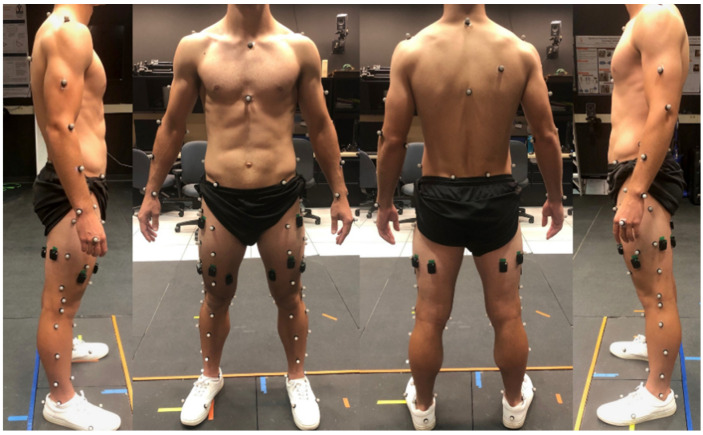
Delsys Trigno sensor and retroreflective motion capture marker placement.

**Figure 2 sensors-21-03622-f002:**
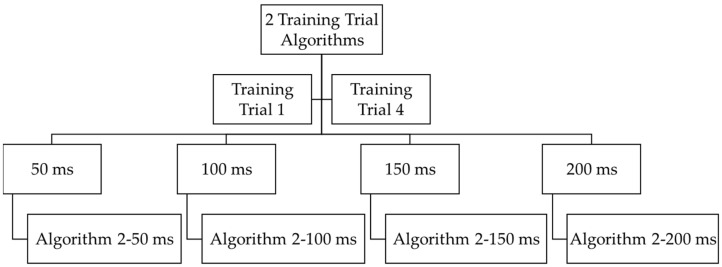
Example of how four algorithms (algorithm 2–50/100/150/200 ms) were trained for one side (right or left) for one subject with two randomly selected training trials (Trial 1 and Trial 4 in this example) out of ten training trials total. This was conducted for both right and left legs for all ten subjects.

**Figure 3 sensors-21-03622-f003:**
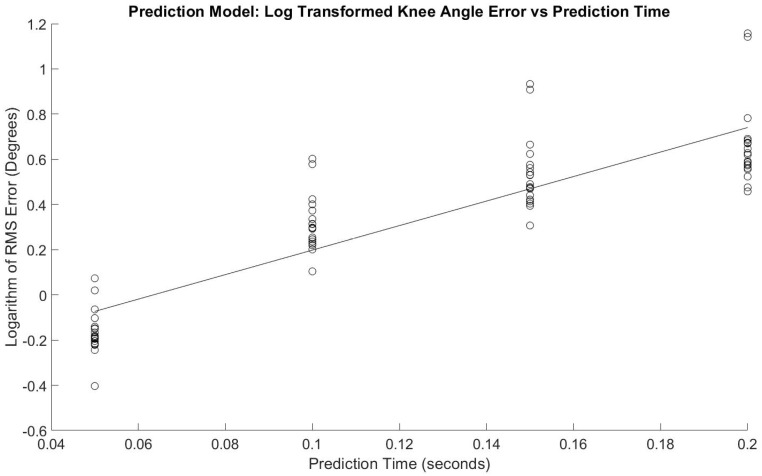
Regression model of logarithmically transformed error for prediction times.

**Figure 4 sensors-21-03622-f004:**
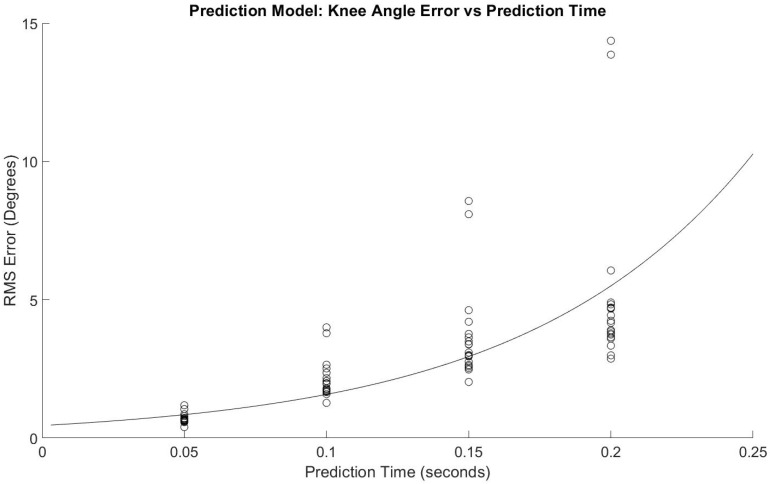
Regression model of error for prediction times.

**Figure 5 sensors-21-03622-f005:**
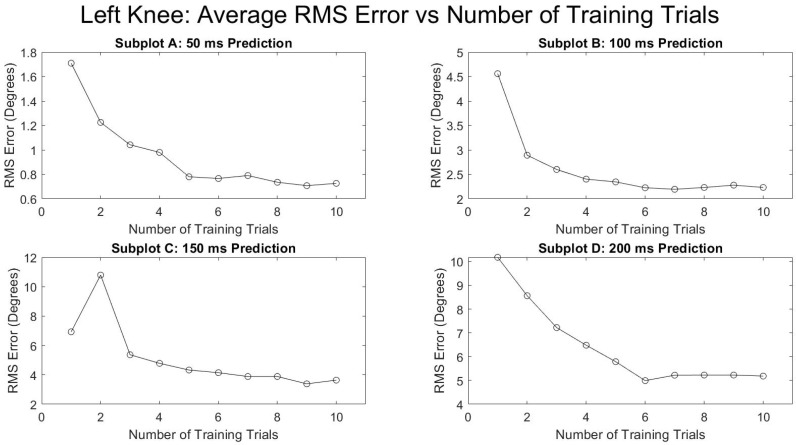
Number of training trials effects on average degrees of RMSE in left knee flexion prediction.

**Figure 6 sensors-21-03622-f006:**
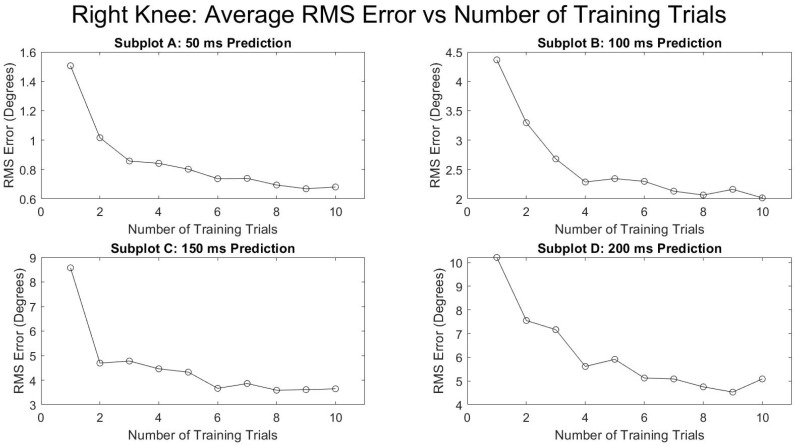
Number of training trials effects on average degrees of RMSE in right knee flexion prediction.

**Figure 7 sensors-21-03622-f007:**
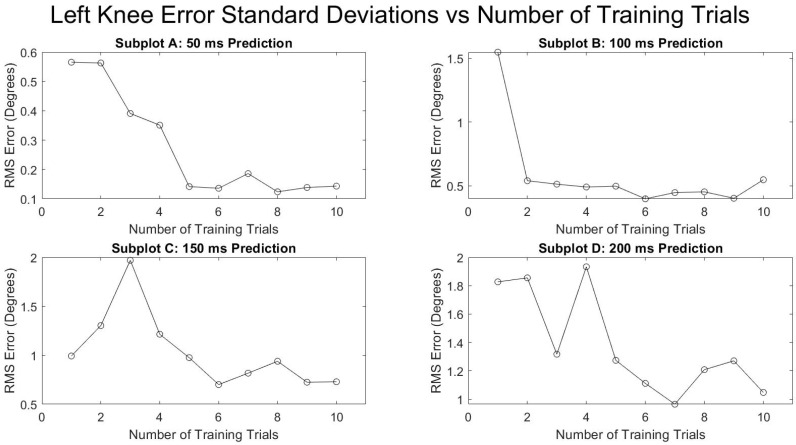
Number of training trials effects on average standard deviation of degrees of RMSE in left knee flexion prediction.

**Figure 8 sensors-21-03622-f008:**
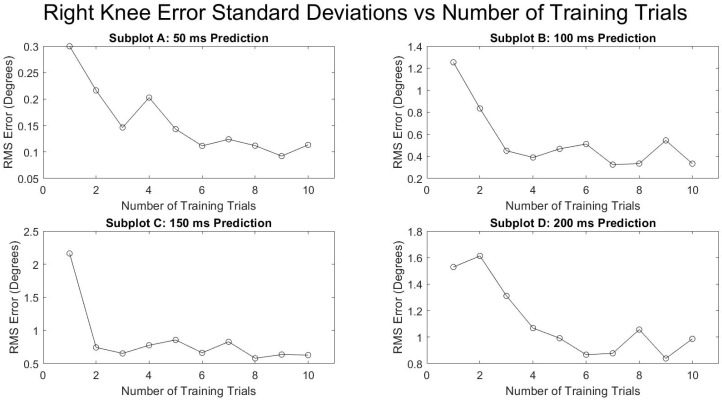
Number of training trials effects on average standard deviation of degrees of RMSE in right knee flexion prediction.

**Figure 9 sensors-21-03622-f009:**
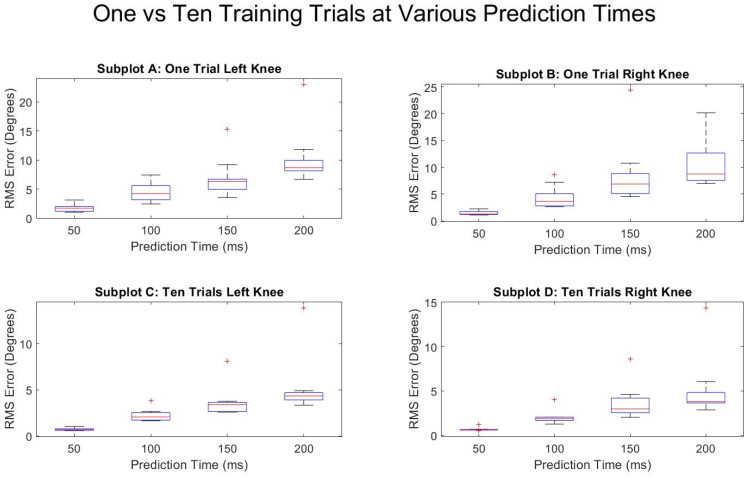
Prediction times effects on average degrees of RMSE in knee flexion prediction for one training trial and for ten training trials. “+” symbol represents outlier data points.

**Table 1 sensors-21-03622-t001:** Analysis of variance results for subject, gender, leg, and prediction time on logarithmically transformed RMSE data.

Source	DF	SS	MS	F	*p*
Sub	4	2.78	0.7		
Gender	1	0.25	0.25	0.46	0.53
Error Sub * Gender	4	2.19	0.55		
Leg	1	0.1	0.1	1.22	0.3
Gender * Leg	1	0.09	0.09	1.08	0.33
Error Sub * Gender * Leg	8	0.67	0.08		
Times	3	42.29	14.1	307.07	<0.001
Gender * Times	3	0.01	0	0.05	0.98
Leg * Times	3	0.03	0.01	0.19	0.9
Gender * Leg * Times	3	0.02	0.01	0.14	0.94
Error Sub * Gender * Leg * Times	48	2.2	0.05		
Total	79				

Note: SS are marginal (type III) sums of squares. “*” represents interaction between variables.

**Table 2 sensors-21-03622-t002:** Tukey HSD pairwise comparisons for times showing significance between each time.

Times	Mean RMSE (°)	Tukey HSD Grouping
50 ms	0.68	A
100 ms	2.04	B
150 ms	3.38	C
200 ms	4.61	D

Error term used: Sub * Gender * Leg * Times, 48 DF. All four means are significantly different from one another.

## Data Availability

The data presented in this study are available on request from the corresponding author. The data are not publicly available due to the human subject-derived nature of the dataset.

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
