# Peer review of "EMG and Joint Angle-Based Machine Learning to Predict Future Joint Angles at the Knee"

_sensors, 2021, doi:10.3390/s21113622_

Round 1

Reviewer 1 Report

The contribution of this work is very good. The title has been formulated unambiguously conveying the focus of the study.  Appropriate research goals are chosen in this contribution, which shows that the authors have a high level of understanding of current research within the field of their research. The authors successfully used the appropriate techniques for analysis of the research objects. The accurate interpretation  of outcomes, well substantiated by the results of the analysis has been achieved by them. The presentation of the results in terms of the research objectives has been successfully made.  The authors have been able to draw logical conclusions from the results.

- Pleasse clarify the training strategy better. In this sense it could useful to put the scheme of the proposed algorithm. For instance you can put a flow chart of the algorithm with some comments.

- Concerning the cited literature the authors can consider the following papers and the literature therein to improve the tutorial/survey aspects of the paper and in particular to inform the reader on different  methods of  prediction in the same context and in the same field field of this publication.

Mercorelli, P., “Biorthogonal wavelet trees in the classification of embedded signal classes for intelligent sensors using machine learning applications”, Journal of Franklin Institute (Elsevier Publishing), vol. 344, no. 6, pp. 813-829, 2007.

Author Response

Reviewer 1:

The contribution of this work is very good. The title has been formulated unambiguously conveying the focus of the study.  Appropriate research goals are chosen in this contribution, which shows that the authors have a high level of understanding of current research within the field of their research. The authors successfully used the appropriate techniques for analysis of the research objects. The accurate interpretation of outcomes, well substantiated by the results of the analysis has been achieved by them. The presentation of the results in terms of the research objectives has been successfully made.  The authors have been able to draw logical conclusions from the results.

Response: The authors would like to thank the Reviewer for the comments above. More specific responses to the Reviewer’s individual comments are listed below.

- Please clarify the training strategy better. In this sense it could useful to put the scheme of the proposed algorithm. For instance you can put a flow chart of the algorithm with some comments.
Response: Thank you for this good suggestion. A figure has been created and added to the manuscript, along with a figure legend, to better explain the training strategy (Figure 2).

- Concerning the cited literature the authors can consider the following papers and the literature therein to improve the tutorial/survey aspects of the paper and in particular to inform the reader on different methods of prediction in the same context and in the same field of this publication.
Response: Thank you for the suggestion. We have gladly included reference to the paper below with relevant corresponding text in the manuscript (Lines 62-63).

Mercorelli, P., “Biorthogonal wavelet trees in the classification of embedded signal classes for intelligent sensors using machine learning applications”, Journal of Franklin Institute (Elsevier Publishing), vol. 344, no. 6, pp. 813-829, 2007.

Reviewer 2 Report

Dear author,

I invite you to increase the introduction by adding a short excursus on all medical applications of EMG in the various fields of medicine; by reporting the difference in reliability of the equipment in the various disciplines as gnatology, orthopedics, physiatry, neurology

these articles could be useful:

·      EMG-Centered Multisensory Based Technologies for Pattern Recognition in Rehabilitation: State of the Art and Challenges PMID: 32722542

·      Facial EMG Correlates of Subjective Hedonic Responses During Food Consumption PMID: 32331423

·      Temporomandibular disc displacement with reduction treated with anterior repositioning splint: A 2-year clinical and magnetic resonance imaging (MRI) follow-up   PMID: 32064850

  • Classification of ankle joint movements based on surface electromyography signals for rehabilitation robot applications.PMID: 27484411

  • A Review of Classification Techniques of EMG Signals during Isotonic and Isometric Contractions. PMID: 27548165

Author Response

Reviewer 2:

Dear author,

I invite you to increase the introduction by adding a short excursus on all medical applications of EMG in the various fields of medicine; by reporting the difference in reliability of the equipment in the various disciplines as gnatology, orthopedics, physiatry, neurology these articles could be useful:

Response: Thank you for the good suggestion. We have gladly included content in the manuscript as suggested, along with references to the literature listed below (Lines 55-57).

  • EMG-Centered Multisensory Based Technologies for Pattern Recognition in Rehabilitation: State of the Art and Challenges PMID: 32722542
  • Facial EMG Correlates of Subjective Hedonic Responses During Food Consumption PMID: 32331423
  • Temporomandibular disc displacement with reduction treated with anterior repositioning splint: A 2-year clinical and magnetic resonance imaging (MRI) follow-up   PMID: 32064850
  • Classification of ankle joint movements based on surface electromyography signals for rehabilitation robot applications.PMID: 27484411
  • A Review of Classification Techniques of EMG Signals during Isotonic and Isometric Contractions. PMID: 27548165

Reviewer 3 Report

This article demonstrates the feasibility of predicting knee flexion angles using ANN for human gait analysis. The study is well presented, and derives critical guidelines on training/testing dataset and the factors affecting overall accuracy. Although the study has innate limitations, they are adequately discussed in the manuscript. 

Author Response

Reviewer 3:

This article demonstrates the feasibility of predicting knee flexion angles using ANN for human gait analysis. The study is well presented, and derives critical guidelines on training/testing dataset and the factors affecting overall accuracy. Although the study has innate limitations, they are adequately discussed in the manuscript. 

Response: The authors would like to thank this Reviewer for the comments above.

Reviewer 4 Report

The paper presents an artificial neural network trained and tested with using knee muscle EMG signals in order to predict knee flexion angles during gait at 50, 100, 150, 11 and 200 ms into the future.

The authors did multiple experiments (ten subjects participated in this study) showing that employing an ANN to predict knee flexion angle within a promising degree of accuracy.

This study is a valuable contribution combining artificial intelligence techniques (learning machines) with experiments performed rigorously and systematically.

The paper is well written, in an appropriate language and presents aspects of an original character accompanied by conclusive experimental results and could be published. 

Author Response

Reviewer 4:

The paper presents an artificial neural network trained and tested with using knee muscle EMG signals in order to predict knee flexion angles during gait at 50, 100, 150, 11 and 200 ms into the future.

The authors did multiple experiments (ten subjects participated in this study) showing that employing an ANN to predict knee flexion angle within a promising degree of accuracy.

This study is a valuable contribution combining artificial intelligence techniques (learning machines) with experiments performed rigorously and systematically.

The paper is well written, in an appropriate language and presents aspects of an original character accompanied by conclusive experimental results and could be published.

Response:  The authors would like to thank this Reviewer for the comments above.